# Mechanical Activation of Granulated Copper Slag and Its Influence on Hydration Heat and Compressive Strength of Blended Cement

**DOI:** 10.3390/ma12050772

**Published:** 2019-03-06

**Authors:** Yan Feng, Jakob Kero, Qixing Yang, Qiusong Chen, Fredrik Engström, Caisa Samuelsson, Chongchong Qi

**Affiliations:** 1School of Resource and Safety Engineering, Central South University, Changsha 410083, China; yan.feng@csu.edu.cn (Y.F.); 21948042@student.uwa.edu.au (C.Q.); 2Division of Minerals and Metallurgical Engineering, Luleå University of Technology, 97187 Luleå, Sweden; jakob.kero@ltu.se (J.K.); fredrik.i.engstrom@ltu.se (F.E.); Caisa.Samuelsson@ltu.se (C.S.); 3Energy School, Xi’an University of Science and Technology, Xi’an 710054, China; qixingyang50@163.com

**Keywords:** mechanical activation, granulated copper slag, blended cement paste, pozzolanic activity, compressive strength, fineness

## Abstract

Mechanical activation of granulated copper slag (GCS) is carried out in the present study for the purposes of enhancing pozzolanic activity for the GCS. A vibration mill mills the GCS for 1, 2, and 3 h to produce samples with specific surface area of 0.67, 1.03 and 1.37 m^2^/g, respectively. The samples are used to replace 30% cement (PC) to get 3 PC-GCS binders. The hydration heat and compressive strength are measured for the binders and derivative thermogravimetric /thermogravimetric analysis (DTG/TGA), Fourier transform infrared spectroscopy (FTIR), and scanning electron microscopy (SEM) are used to characterize the paste samples. It is shown that cumulative heat and compressive strength at different ages of hydration and curing, respectively, are higher for the binders blending the GCS milled for a longer time. The compressive strength after 90 d of curing for the binder with the longest milling time reaches 35.7 MPa, which is higher than the strength of other binders and close to the strength value of 39.3 MPa obtained by the PC pastes. The percentage of fixed lime by the binder pastes at 28 days is correlated with the degree of pozzolanic reaction and strength development. The percentage is higher for the binder blending the GCS with longer milling time and higher specific surface area. The pastes with binders blending the GCS of specific surface area of 0.67 and 1.37 m^2^/g fix lime of 15.20 and 21.15%, respectively. These results together with results from X-ray diffraction (XRD), FTIR, and SEM investigations demonstrate that the mechanical activation via vibratory milling is an effective method to enhance the pozzolanic activity and the extent for cement substitution by the GCS as a suitable supplementary cementitious material (SCM).

## 1. Introduction

Copper slag (CS) is a by-product generated in the process of copper smelting and converting [1]. It has been estimated that for every ton of copper produced, there are about 2.2–3 tons of CS generated [2], resulting in an annual generation rate for the slag of approximately 40 million tons worldwide [3]. Over recent decades, substantial effort has been invested for reusing this slag in the construction industry to decrease the slag quantity being deposited. The effort has led to uses of the CS as coarse or fine aggregates with favorable physical and mechanical properties over a large application area. The area comprised mainly road-base construction, railroad ballast, asphalt pavements, and mine backfill, as well as the cement and concrete production [3,4,5]. Some researchers also studied pozzolanic behaviors of the CS and revealed possibilities for its applications in cement and concrete industry as supplementary cementitious materials (SCM) [6,7]. The benefits of using CS in cement included improvements in mechanical strength and durability, reduction in hydration heat, etc. 

Granulated copper slag (GCS), produced by water granulation of liquid CS, consists generally of SiO_2_, FeO, and minor amounts of other compounds, with complex silicate amorphous phase formed due to the rapid water cooling [8]. The GCS, with high contents of SiO_2_ and amorphous phase, possesses a potential of taking part in pozzolanic reactions, while it is exposed to alkaline environment [9,10]. In the environment, the glassy network in GCS can be decomposed to release silicate species, which react with calcium hydroxide (CH) to form calcium silicate hydrates (C–S–H) [11]. The alkaline environment and CH for the reactions can be produced by cement hydration. The C–S–H formed from the reactions, with its C/S ratio lower than the one for the C–S–H produced by cement hydration, may thus be referred to as pozzolanic C–S–H. Both types of the C–S–H contribute to the strength development in cement products. 

Rojas et al. [6] reported the pozzolanic reactivity of GCS being rather similar to that of fly ash (FA). The GCS thus exhibited a low reaction rate, when being used to partially replace cement, and could only contribute to the development of compressive strength for mortar at late ages. This has become a major obstacle to the use of GCS as a pozzolanic material and made it necessary to conduct new research work for enhancing the reaction rate, to increase the extent of cement substitution by GCS. 

Literatures show that mechanical activation via high-energy milling is an efficient way to enhance the reactivity for industry by-products and pozzolanic materials, such as granulated blast furnace slag (GBFS), steelmaking slag and FA [12,13,14,15]. The milling treatments have improved mechanical strength and durability for the binders formed by blending the treated materials in cement. The reactivity enhancement for these materials is attributable to combined effects of surface area, structural disorder, surface physicochemical properties, and other defects induced by the milling. There is hardly any literature information regarding mechanical activation of GCS. It is also difficult to use the results described above about the activation of the other by-products. As the chemical composition (contents of Ca and Fe oxides) and formation temperature of the GCS are quite different from these of the GBFS, steel slag and FA [14]. 

A mechanical activation of GCS with vibratory milling is thus carried out in the present investigation, by referring to the results from the early studies [12,13,14]. Effects of the GCS activation on the pozzolanic activity and microstructure of hydrated samples are investigated. The study results are reported and discussed in this article to aid enhancement of use for the GCS as a SCM in cement. The enhanced use can reduce the GCS deposition and, furthermore, decrease not only consumptions of energy and natural raw materials, but also emissions of greenhouse gases related to the manufacturing of cement clinker, thus enhancing sustainability for both copper and cement industry.

## 2. Materials and Methods

The major materials used in this investigation are GCS obtained from a copper smelter in Sweden and Byggcement (CEM II/A-LL 42.5 R) coded as PC, which chemical compositions were determined using X-ray fluorescence spectroscopy (XRF, Bruker AXS, Karlsruhe, Germany) and reported in Table 1. 

The received granular GCS was ground with a rod mill for 15 min and was sieved, after the grinding, using standard sieves to obtain a feedstock powder with diameters ranging from 0.075 to 0.60 mm. The mechanical activation was performed using a vibrating mill (Humboldt Palla 20U, Humboldt, Germany) with porcelain cylindrical grinding media of 10 mm diameter. The amplitude and frequency of vibrating for the mill were 10 mm and 1000 rpm, respectively. The feedstock powder was treated by the mill for 3 different time intervals, 1, 2, and 3 h, to determine the effects of the milling time on the degree of mechanical activation. The weight of the powder for each of the milling treatments was 1 kg. 

The size distribution of the particles with the size range of 0.04–500 μm was determined by laser diffraction using a particle size analyzer (CILAS 1064, Marseille, Cilas, France). The BET (Brunauer–Emmett–Teller) surface area was measured using a Micromeritics FlowSorb II 2300 instrument (Micromeritics, Norcross, GA, US) by adsorption of nitrogen. X-ray diffraction (XRD, Panalytical Empyrean, Almelo, The Netherlands) was employed to investigate changes in the mineralogy and contents of amorphous phases in the GCS powder milled for different durations. Diffraction patterns were measured in 2θ range of 10–70° using Cu-Kα radiation of 40kV and 40 mA and at a step size of 0.026°.

In accordance with Chinese national standard GB/T 12957-2005, the GCS powder mechanically activated by milling for 1 h, 2 h, and 3 h was used to blend in the cement, with a cement replacement ratio of 30%, to prepare three binders coded as CS1, CS2, and CS3, respectively. The hydration heat at early ages was continuously monitored by isothermal conduction calorimeter (TAM Air device, TA Instruments, New Castle, Delaware, US) at 25 °C within 120 h. These measurements were performed for paste samples prepared using the binders, CS1, CS2, and CS3, and pure cement (PC) with a water-to-binder ratio (W/B) of 0.5. Mixtures of the paste samples were also prepared and cast into right-prism molds of dimensions 4 × 4 × 4 cm^3^. The de-mold was performed at 24 h after the casting and paste cubes were obtained and stored under laboratory conditions (20 ± 2 °C and relative humidity 100%). The compressive strengths of the cubes were tested at 7, 28, and 90 days. 

The samples of some cubes cured for 28 days were dried in an oven at 50 °C for 3 days to stop hydration. The dried samples were analyzed by DTA/thermogravimetric analysis (TGA) to obtain weight change results for calculating contents of CH and calcium silicate hydrate (C–S–H). The TGA and derivative thermogravimetric analysis (DTG) were performed using a STA 449C apparatus (Netzsch, Germany), with samples heated with the rate of 10 °C/min in the temperature range of 20 to 900 °C and protected by a N_2_ flow rate of 100 ml/min. FTIR (Thermo Electron, Madison, WI, US) spectra of the dried powder samples were recorded on a Nicolet Nexus 470 spectrometer with KBr wafer in the wavenumber range 4000–400 cm^−1^. Microstructural characterization of the paste samples was carried out using scanning electron microscopy (SEM, Zeiss Merlin, Oberkochen, Germany), with the analysis performed on gold plated polished sample surfaces using an accelerating voltage of 5 kV. The purposes for these analyses were to determine extents of the pozzolanic reactions for the GCS treated by the mechanical activation. 

## 3. Results and Discussion

### 3.1. Mineralogy and Particle Size Distribution of the Milled GCS

The XRD patterns of the feedstock and the GCS treated with the different milling durations are presented in Figure 1. Each of the patterns shows the diffuse scattering feature of amorphous structure with the broad diffuse hump peak between 20° and 40°, which is a typical characteristic of vitreous structure in the GCS. Rojas et al. [6] reported the XRD spectrum of GCS with the specific surface area of 0.68 m^2^/g, which is similar to the XRD spectra seen in Figure 1. Similar patterns were also reported in literature for some other silicate materials with existences of glassy phase [16,17]. The high content of amorphous phase is related to a high degree of vitrification of GCS due to the rapid water cooling, which can potentially provide sufficient reactive SiO_2_ for the pozzolanic reaction. 

The XRD pattern shows the presences of fayalite and magnetite in feedstock powder (<0.6 mm) with the peaks identified at 25.0°, 31.6°, 34.9°, 51.3° and 35.8°, respectively. Diffraction peaks corresponding to crystalline phases are hardly detected for the 3 samples after the vibratory milling, suggesting that a complete vitrification of the GCS powder was achieved after the first hour of milling. Therefore, the GCS samples milled over 1 h can be considered as glassy materials, in which the amorphous phase is mainly composed of polymerized network of silicate tetrahedral [11]. 

Figure 2 shows the cumulative particle size distribution of the GCS samples. With the mechanical activation, a comminution of GCS particles occurred inducing changes in particle size distribution and specific surface area (BET). The treatment with vibratory milling results in a reduction of the particle size for the GCS and the figure shows a shift of the cumulative distribution curve towards the smaller diameters, demonstrating a dependence of the size distribution on the milling durations. The most significant size reduction was achieved in the first hour of milling, by which the median size (D50) decreased from 152.2 to 9.4 μm. For a further increase in milling time, the reduction effect appears to be weakened with the size of the particles distributed over a much narrower range. 

The characteristic particle sizes (D10, D50 and D90) and specific surface area of GCS are seen in Table 2. Values of D90 are 27.2 and 14.2 μm after the milling of 1 and 3 h, respectively, demonstrating a steep decrease in the average diameter for coarse particles up to 3 h of vibratory milling. The similar variations are also obtained for the lower classes (D10 and D50), which extents of the variations are however smaller than that for D90. This suggests that the vibratory milling is more efficient in the case of treating larger particles. 

Bouaziz et al. [15] determined the optimal milling time of 1 h or less for small particles (<30.0 μm) in GBFS with a planetary high-energy ball mill, and found that the agglomeration could occur at a longer milling time. The different trend can be explained by the differences in types of materials and milling. Results of FA milling reported by Hamzaoui et al. [18] show that the fracture of particles corresponding to all size classes develops within the first 3 h, followed by agglomeration as milling continues. While milled for the time up to 3 h, the tested FA exhibits a behavior similar to GCS in the present study. Besides, no evidence of agglomeration for the GCS particles of different size classes is attained from the measurements of specific surface area. As seen from Table 2, the specific surface area of GCS increased from 0.67 to 1.37 m^2^/g by prolonging the time for vibratory milling from 1 to 3 h, respectively. Higher levels of fineness (0.67–1.37 m^2^/g) are achieved for the present GCS than these (0.27–0.40 m^2^/g) for other materials reported in [6,7,19]. The higher fineness levels may benefit the pozzolanic activity for the GCS, especially at the early ages [20]. 

Tsuyuki et al. [21] confirmed effects of mechanical activation on enhancing specific surface area and on consequent improvement for the reactivity. The authors also found that the particle fracture can increase the concentrations of ionic types of glass phase in the surface of GBFS particles, contributing to the evolution of hydration. According to the modified random network (MRN) model for glass structure proposed by Greaves [22], network modifiers and non-bridge oxygens (NBOs) exhibit a non-random and inhomogeneous distribution throughout the amorphous structure of GCS, leading to separated rich regions of modifiers. The agglomeration of weak ionic bonds in modifier-rich regions may contribute to the formation of weak interfaces, where GCS particle breakage can preferentially occur during the milling process [21]. The selective fracture of particles enriches the distribution of ionic bonds on freshly exposed surfaces and this enrichment gradually increases with increasing milling duration. 

Since the ionic bonds are more reactive than the covalent bonds during the decomposition of glassy structure [11], the mechanical activation can produce more reactive ionic bonds to favor network depolymerization, which can also improve the pozzolanic reactivity of GCS. Thus, the contribution to enhancement of pozzolanic activity of GCS could be attributed not only to the increase of the surface area available for reacting, but also to the modification of the chemical structure of the particle surface, as the mechanical activation by vibratory milling may induced both effects. It may be needed to conduct some new work in near future for a better clarification of the effect of modification for the chemical structure of the particle surface resulted from the mechanical activation of the GCS and other waste materials. 

### 3.2. Isothermal Calorimetry

To investigate the influence of mechanically activated GCS on the early hydration, isothermal calorimetry is used to continuously monitor the hydration heat development of blended binders CS1, CS2, and CS3 within 120 h. Results of normalized heat flow and cumulative heat are shown in Figure 3. The profile of PC is also given as a reference. Characteristic heat values are presented in Table 3.

There are five stages of heat development for a typical Portland cement, the initial period (I), the induction period (II) the acceleration period (III), the deceleration period (IV) and period of slow continued reaction (V) [23]. The initial period of the PC hydration (I) is characterized by a strong and transient peak of heat flow within the first few minutes (Figure 3a). This early exothermic signal is mainly attributed to the wetting of cement powders and the rapid dissolution of the tricalcium silicate (C_3_S) in the powder followed by a sharp reduction in dissolution rate [24]. Some authors [25,26] proposed that the deceleration is caused by the rapid formation of intermediate silicate hydrate phase on the C_3_S particles. The hydrate phase acts as a metastable and protective layer to restrict access of the particles to water. Other authors [27,28] favor the hypothesis that the rapid dissolution ends with the formation of a superficially hydroxylated layer on the C_3_S surface in contact with water followed by a congruent dissolution. Consequently, the dissolution rate of C_3_S decelerates very quickly with an increase in the concentration of CH. The hydrated phase (C–S–H) nucleates rapidly on C_3_S surface from the supersaturated solution then grows slowly. This is followed by the induction period (II) with an extremely low rate of heat release, during which a dynamic equilibrium is gradually reached between the slow dissolution of C_3_S and the initial growth of C–S–H [24]. After the brief period, the acceleration period (III) comes at 2.3 h with a strong exothermic signal produced by the hydration of C_3_S that becomes dominated by the rapid and massive nucleation and growth of the C–S–H and CH. The heat flow increases significantly and reaches a maximum corresponding to the maximum hydration rate at the end of the acceleration period (at about 8.0 h). A sharp decrease in the hydration rate occurs subsequently during the deceleration period (IV) owing to the transition of hydration process from chemical controlled to diffusion controlled [29]. The diffusion of reactants slows with the formation and growth of thick hydrated layer [24]. During the slow continued reaction period (V) the hydration finally reaches a very low rate and develops continuously with a low heat output.

According to the investigations in other pozzolanic materials [30,31], the substitution of cement by GCS affects the hydration of cement over different stages owing to dilution and physical and chemical effects. As shown in Table 3, the induction period of hydration for PC ends after 2.3 h, which is prolonged to 2.5–2.8 h by addition of mechanically activated GCS. The retardation is mainly related to the dilution effect of GCS [30]. The incorporation of GCS into the binders increases water-to-cement ratios (w/c) and reduces the concentrations of calcium and silicate in the pore solution. The time required for solution to reach a supersaturated state becomes longer due to the incorporation that also delays the induction period. 

Besides, the curves in Figure 3a show a gradual decrease in the degree of retardation with increasing GCS fineness. This can be explained by acceleration compensation from the physical (filler) effect of GCS. The GCS with a longer time for mechanical activation possess larger surface area, Table 2, and can provide additional and efficient nucleation sites for the early precipitation and growth of C–S–H, which accelerates the cement hydration and compensates for the delay for the induction period. The ending time for the induction period is thus decreased from 2.8 h for binder CS1 blending GCS milled for 1 h to 2.6 and 2.5 h for binders CS2 and CS3 blending CGS milled for 2 and 3 h, respectively, Table 3. 

A similar trend was observed for the maximum of heat flow during the acceleration period, providing further evidence for the physical effect. The maximum values for the second exothermic peaks of the binders occur later than the PC. Here, the dilution effect still dominates during the acceleration period, resulting in lower peak values for the binders than that for PC. The peak values for the binders are influenced by the milling time and fineness for the GCS, which increases from 2.2 to 2.3 mW/g with increasing GCS fineness. 

Unlike a single exothermic peak for PC, there are another exothermic peak appears in the later period on the heat flow curve for the binders, owing to the chemical effect of GCS [31]. The chemical effect is mainly ascribed to the pozzolanic reaction of GCS, in which GCS reacts with CH generated by the cement hydration to form pozzolanic C–S–H. Due to the slow pozzolanic reaction, the chemical effect plays an important role after the acceleration period, during which a large amount of CH is produced to provide alkaline environment for the decomposition of glassy particles in GCS. Increases of values for the third exothermic peaks from 2.3 to 2.4 mW/g, Table 3, indicates an improvement in pozzolanic activity achieved by the increase of GCS fineness due to the milling treatments. 

The heat flow for both PC and the binders finally decreases significantly and reaches a minimum during the period of slow continued reaction. The GCS contributes largely to the hydration heat in the period via the pozzolanic reaction, which is reflected by the higher heat flow for the binders than that for the PC after the time of 60 h (Figure 3a). 

Figure 3b and Table 3 show the cumulative heat curve in period of the first 120 h and total heat emission at 12, 24, 48, and 120 h for PC and the binders. The cumulative heat increases rapidly within 40 h, then the rate slows down appreciably, matching the heat flow results (Figure 3a). This indicates that the hydration process of the binders was dominated by the cement hydration in the early period. The binders containing GCS exhibit a substantial decrease in the cumulative heat, especially after 24 h, owing to the cement dilution effect. 

The data in Table 3 show that among the 3 binders, values of the cumulative heat at different ages of hydration are higher for the binders blending the CGS milled for a longer time, especially in the later periods. This can be attributed mainly to the enhanced pozzolanic activity of the GCS by increasing the intervals of the mechanical activation and the matched fineness.

It can be seen from Table 3 that the total heat emission at 120 h for PC is 219.0 J/g, which is higher than these for the binders. The values for the binders, varying from 175.8 to 183.9 J/g, are greater than 70% of the heat emission for PC (153.3 J/g). Similar results can also be observed for other periods, in which the total heat emission of the binders exceed the amount of the heat released independently by the hydration of cement (70%) in these binders. The extra heat is mainly produced by the pozzolanic reaction between the mechanically activated GCS and CH generated by cement hydration. On the other hand, the consumption of CH by pozzolanic reaction promotes also the cement hydration, as proposed by Han et al. [32]. 

### 3.3. Compressive Strength

The results of compressive strength of the pastes prepared by using PC and the binders are shown in Figure 4. The ratio of the strengths for binder pastes to that for the PC paste and the strength development rate in different period are calculated with the results presented in Table 4.

A retardation of early development for the compressive strength is observed for the binder pastes when compared with the PC paste. Especially, noticeable is the case for paste with blending the GCS after vibratory milling for 1 h (CS1). As shown in Figure 4, the CS1 paste gained compressive strength of 14.0 MPa at 7 d, which is less than 50% of the strength at the age for the PC pastes. This negative effect on the early-age strength can be explained by the cement dilution effect, which can be mitigated by prolonging the curing time. The compressive strengths at 28 and 90 d for CS1 are 56 and 72% of that for the PC paste (Table 4), respectively. This attenuation of dilution effect on the strength development can also be observed for other binder pastes (CS2 and CS3). 

Although the binder pastes show low strength values at early ages, their strength gain rates are higher than these for the PC pastes in the curing period of 7–28 and 28–90 d. This is mainly attributed to the additional C–S–H gel formed through the pozzolanic reaction between the GCS activated mechanically and CH generated by PC hydration. The pozzolanic reaction at early ages plays a secondary role in the development of strength, which is insufficient to compensate for the dilution effect. The development of compressive strength is gradually dominated by the pozzolanic reaction with curing time, resulting in a sharp increase in the strength values at later ages for the binder pastes.

The binder pastes of different curing ages show a significant increase in compressive strength with increased duration of mechanical activation for the GCS. This indicates that the milling duration can influence significantly on the mechanical strength of pastes by increasing the fineness of GCS. The similar results were also obtained in the experiments on other amorphous materials, such as GBFS [33] and FA [34]. These studies have shown that the reactivity of these materials can be enhanced by increasing the duration of mechanical activation. This enhanced reactivity, benefiting the development of mechanical strength of cement products, is facilitated by the combined effect of fineness and surface structure. In the present study, the highest activity is achieved for the sample of the GCS with the highest fineness or specific surface area of 1.37 m^2^/g, resulted from the longest milling duration of 3 h. The pastes of CS3 (blending in the 3h-milled-GCS) therefore acquire higher strength values than these for CS1 (blending in the 1h-milled-GCS) at 7, 28, and 90 d by approximately 35, 50 and 25%, respectively. The compressive strength of CS3 after 90 d of curing reaches to 35.7 MPa, close to 39.3 MPa, the strength value obtained by the PC pastes, Figure 3. 

Table 4 shows that the strength development rate for the binder pastes is significantly affected by the curing period and fineness of GCS. The paste samples of CS2 and CS3 have higher rates (0.40–0.43 MPa/d) of strength development in the period from 7 to 28 d than that (0.26 MPa/d) for the PC pastes. The rate (0.21 MPa/d) is the lowest for the CS1 pastes due to the low rate for the pozzolanic reaction related to the milling in the shortest time of 1 h for the GCS. 

Nevertheless, in the period of 28–90 d, the binder pastes develop their strength with rather low rates (0.10–0.16 MPa/d), close to 0.1 Mpa/d gained by the PC paste. It is estimated that the pozzolanic reaction of mechanically activated GCS occurs mainly during 7–28 d to contribute to the high-rate strength development for the binder pastes. By increasing GCS fineness, the pozzolanic activity of GCS is enhanced to accelerate the development of compressive strength, which is in good agreement with the results of hydration heat evolution reported in Section 3.2. With the further development of pozzolanic reactions, CH and GCS particles are largely consumed, causing a reduction in the pH value of the pore solutions and the number of reactants, thus limiting the extent for further reaction and slowing down the strength development rate in the later period (28–90 d). 

### 3.4. Thermal Analysis

The DTG/TGA results of the samples from pastes after curing for 28 days are presented in Figure 5. The pastes were prepared using binders (cement PC + 30% of GCS activated mechanically) and PC. DTG curves in Figure 5a show three endothermic peaks recorded for these pastes. The first endothermic peak recorded between 50–200 °C is due mainly to the dehydration of C–S–H and decomposition of ettringite (AFt) [35]. The second peak observed in the temperature range of 400–500 °C corresponds to the dehydration of CH [36]. The last endothermic peak appearing at about 670 °C is attributable to the decarbonation of calcium carbonate (CaCO_3_) [37]. 

The presence of C–S–H in the current study can be identified for all the paste samples by two endothermic signals in the range of 50–200 °C. The first signal is an endothermic peak at about 80 °C, corresponding to the decomposition of C–S–H (I) [38], the intensity of which increases with increasing the duration of mechanical activation for the GCS, especially for the peak on the top curve in Figure 5a for CS3 blending the GCS milled for 3 h. The second signal is a shoulder at about 120 °C, superposed on the right side of the first peak, and indicates the presence of C–S–H (II) and ettringite (AFt) [37], which intensity is also dependent on the milling duration of the GCS in the way rather similar to the peak near 80 °C. The shoulder intensity becomes too weak to be distinguished on the top curve in Figure 5a, when the fineness of GCS is increased up to 1.37 m^2^/g by the 3 h milling. This is in good agreement with the model proposed by Taylor [39], who assumed that the C–S–H phase formed during the C_3_S hydration at ambient temperature consists of tobermorite-type and jennite-type structures, i.e., C–S–H (I) and C–S–H (II), respectively. The mechanically activated GCS, possessing greater fineness and activity, tends to consume larger amounts of CH in the pozzolanic reaction, leading to a decrease of Ca/Si ratio in the system. The transformation from C–S–H (I) to C–S–H (II) can thus be inhibited or delayed [10], which results in the intensity reduction of C–S–H (II) observed on the DTG curves. 

It is noticeable that no obvious evidence for the presence of monosulphate (AFm) can be found for the binder pastes in the temperature range of 50–200 °C. Similar results were also obtained from the experiments on the blended cement prepared with CaCO_3_ or GBFS [38], which was attributed to the absence of AFm formation.

The weight loss data of TGA corresponding to the endothermic peaks on the DTG curves can be employed to determine the quantity of hydration products in the pastes. Considering that the weight loss recorded between 50 and 200 °C is attributable largely to the dehydration of C–S–H, the quantity of C–S–H formed in each of the samples of 28-day curing can then be deduced from the weight of interlayer water. The TGA curves in Figure 5b indicate lower quantities of C–S–H in the binder pastes as compared with that in PC paste because of the cement dilution. 

It is clear, while a compression is made among the binders, that increasing the fineness of GCS favors the formation of more C–S–H gel in the binder pastes, as well as a higher compressive strength. This is consistent with the increase in the intensity of the first endothermic peak on the DTG curves. A maximum weight loss of 9.27% is measured for the paste prepared using binder CS3 containing the GCS with the highest fineness of 1.37 m^2^/g. The loss of 9.27% is close to that for the PC paste (10.11%) and is indicative of a formation of C–S–H gel with a maximum quantity (among the 3 binders) to develop the highest compressive strength of 27.9 Mpa at 28 d for binder paste CS3.

The other two weight losses are in temperature ranges of 400–500°C and 600–800 °C associated mainly with the dehydration of CH and the decarbonation of CaCO_3_, respectively. The difference between the PC and the binder pastes in the total quantity of CH, including both carbonated and non-carbonated CH [40], can be related to the degree of pozzolanic reaction of GCS. 

The quantity of CH produced and afterwards consumed in the paste of cement blending a SCM can be determined at a certain age to monitor the evolution of the pozzolanic reaction of the SCM. This is also indicative of the rate at which the SCM releases its reactive constituents from its glassy network decomposing into pore solution towards fixing available lime. The method described in [41] is thus employed to compute the percentages of fixed lime in the binder pastes. 

Values of the fixed lime calculated for pastes CS1, CS2 and CS3 are 15.20, 17.25 and 21.15%, respectively. The lowest value of fixed lime for CS1 paste corresponds to the lowest rate at which the blended GCS with the fineness of 0.67 m^2^/g releases its reactive SiO_2_ into solution to bind the lime, suggesting a slower pozzolanic reaction. This is in good agreement with the results reported by Rojas et al. [6]. The GCS with greater fineness increases the fixed lime value more effectively, consistent with the trend for changes of C–S–H quantity, resulted from the enhancement of pozzolanic reaction to produce larger quantities of C–S–H in the pastes for the strength enhancement.

The pastes CS1 and CS3 blending with the GCS of specific surface area of 0.67 and 1.37 m^2^/g fix lime of 15.20 and 21.15%, respectively. These show that the percentage of fixed lime for the binder pastes at 28 days increases with increasing GCS fineness or the specific surface area, but the increase rate for the fixed lime is lower than the increase rate for the paste strength, cf. Figure 3. This difference is mainly due to the packing (physical) effect of fine grains in the GCS [42,43], which can fill the voids of the paste, resulting in denser packing between the mineral particles and the paste and contributing to the improvement of compressive strength. The enhancement of compressive strength for binder pastes by mechanical activation with vibratory milling can thus be attributed to not only the acceleration of pozzolanic reaction, but also the packing effect of finer GCS particles.

### 3.5. FTIR Spectra Analysis

Figure 6 presents the FTIR spectra of the cement-GCS pastes for 28 days of curing. The similarity in the spectra for all measured samples show that reaction products formed in the mixtures containing GCS with different fineness are rather similar. The absorption peak observed at about 3620 cm^−1^ is resulted from the bending vibration of OH of Portlandite (CH) [44], which is formed with the hydration reaction of cement. The intensity of this peak decreases slightly with increasing GCS fineness, indicating a larger consumption of CH and consequently the achievement of higher degrees of pozzolanic reaction. The prominent peaks located at 3440 and 1640 cm^−1^ in each spectrum are due to the asymmetric stretching and bending vibrations, respectively, of the OH group in bound water [45]. The absorption peaks at wavenumber of 1420 cm^−1^ and 870 cm^−1^ corresponds to the asymmetric stretching of C–O bond of the CO_3_^2−^ group in calcite (CaCO_3_) [46], which is introduced into the pastes with the carbonation of CH. The carbonation occurs for all the three samples and the degree is increased with increasing GCS fineness, as expected, due to the decreased residual CH in the pastes. The variation of CH and CaCO_3_ in the FTIR spectra is consistent with that observed in the DTG curves. The peak at about 1100 cm^−1^ is attributed to the asymmetric stretching vibration of S–O bonds of SO_4_^2−^ groups in AFt [47], which is formed during the reaction between calcium aluminates and gypsum. 

The peak at about 970 cm^−1^ is attributed to Si–O stretching vibration of SiO_4_ tetrahedron in the formed C–S–H gel [44]. The C–S–H gel in the binder pastes is generated not only by hydration reaction of PC, but also the pozzolanic reaction between mechanically activated GCS and CH, in the presence of water. For the pastes with greater fineness GCS, the peak corresponding to C–S–H becomes more prominent whereas the peak of CH further decreases in intensity, demonstrating the effect of GCS with a high degree of fineness on the enhancement of pozzolanic reaction. Moreover, a small shift in this peak towards lower wavenumbers can be observed, which appears at 973, 971 and 970 cm^−1^ for the samples of CS1, CS2 and CS3, respectively. This shift may be related to the decrease in Ca/Si ratio of C–S–H due to the proportion of different types of C–S–H, as mentioned in Section 3.4, C–S–H (I) and C–S–H (II). 

The results of FTIR studies are consistent with the DTG/TGA measurements of the reaction products in the PC-GCS system. The GCS with greater fineness and activity favors the increase in the quantity of C–S–H gel by consuming CH. This can be used to explain the enhancement of compressive strength for the binder pastes via enhancing the GCS fineness by the present mechanical activation. 

### 3.6. Microstructural Analysis

SEM micrographs of hydration products in the binder pastes at 28 days are shown in Figure 7. A mixture of hydration products, unreacted GCS particles and pores is observed in the micrographs for each of the samples. The FTIR results indicate that C–S–H, CH and AFt are the main hydration products in the pastes. As shown in Figure 7a, small quantities of the flocculent (gel-like) products formed from the pozzolanic reaction are attached to the surface of unreacted GCS particles, suggesting a low degree of decomposition and reaction for the GCS milled for 1 h. The pores have been partially filled with the unreacted GCS particles and needle-like hydrates of AFt. 

Figure 7b,c show fewer unreacted GCS particles existing in CS2 and CS3, where the gel-like hydrates grow into coarse flaky clusters with compact microstructure. This demonstrates that the GCS with higher surface areas exhibits an enhanced pozzolanic reactivity, which is consistent with the results reported in Section 3.1, Section 3.2, Section 3.3, Section 3.4 and Section 3.5. As expected, no hexagonal plates of CH crystal are observed due to its consumption during the pozzolanic reaction [48], while only a few CH with irregular shapes are detected in the binder pastes. In addition, with an increasing degree of mechanical activation of GCS, more fibroid hydrates around the gel-like hydrates are observed. The hydrates can fill the pores and aid the hydrate binding, benefiting the strength development for the pastes.

### 3.7. Suggestions of Furture Work

It is suggested to conduct some new work in near future for a better clarification of the effect of modification for the chemical structure of the particle surface induced by the mechanical activation of the GCS and other waste materials using different milling methods and equipment. 

Further experiments with larger scales may be carried out to determine the optimal milling duration of GCS and substitution ratio of cement for the industrial application. One of the applications may be the cemented paste backfill [49]. The mixtures of GCS and wastes from other industry may also be used for studies of mechanical activation.

## 4. Conclusions

GCS possesses a low pozzolanic activity, which limits its substitution rate in cement as a SCM. The presesnt study, mechanical activation of the GCS, is then performed to increase the substitution rate for the GCS as SCM, as well as sustainability for both copper and cement industry. The GCS is milled in a vibration mill with different durations (1, 2, 3 h) to obtain 3 samples, which are used to replace 30% cement (PC) forming 3 PC-GCS binders coded as CS1, CS2 and CS3. The hydration heat and compressive strength are measured for the binders. DTG/TGA, FTIR and SEM are used for characterizations of paste samples. The results from the measurements and characterizations are summarized below:

(1) Vibratory milling proves to be an effective way to achieve high fineness of GCS with no agglomeration occurring within 3 h of milling. 

(2) The cumulative heat measured for the binder CS3 is 183.9 J/g, which is greater than 70% of the heat emission for PC (153.3 J/g) and greater than other two binders with GCS milled with shorter time. 

(3) The pastes of CS3 acquire higher strength values than these for CS1 and CS2 (blending in GCS milled with shorter time) at 7, 28 and 90. The compressive strength of CS3 after 90 d of curing reaches to 35.7 MPa, which is higher than the strength of other binders and close to the strength value of 39.3 MPa obtained by the PC pastes. 

(4) The percentage of fixed lime for the binder pastes at 28 days is related to the degree of pozzolanic reaction and strength development. The percentage is higher for the binder blending the GCS with longer milling time and higher specific surface area. The pastes CS1 and CS3 blending the GCS of specific surface area of 0.67 and 1.37 m^2^/g fix lime of 15.20 and 21.15%, respectively.

(5) The peak of FTIR spectra located at about 970 cm^−1^ is attributed to Si–O stretching vibration of SiO_4_ tetrahedron in the formed C–S–H gel. For the pastes with GCS of greater fineness, the peak corresponding to C–S–H becomes more prominent, demonstrating the effect of GCS with a higher degree of fineness on the enhancement of pozzolanic reaction. 

(6) No hexagonal plates of CH crystal can be observed in SEM micrographs of CS2 and CS3 due to its consumption by the pozzolanic reaction. With an increase of the degree for mechanical activation of GCS, more fibroid hydrates around the gel-like hydrates are observed. The hydrates can fill the pores and aid the hydrate binding, enhancing the strength development for the pastes. 

These results confirm the effectiveness of mechanical activation via vibratory milling up 3 h as a viable method of enhancing the pozzolanic activity of GCS, which can be adopted to increase the rate for cement substitution by the GCS as a high quality SCM. 

## Figures and Tables

**Figure 1 materials-12-00772-f001:**
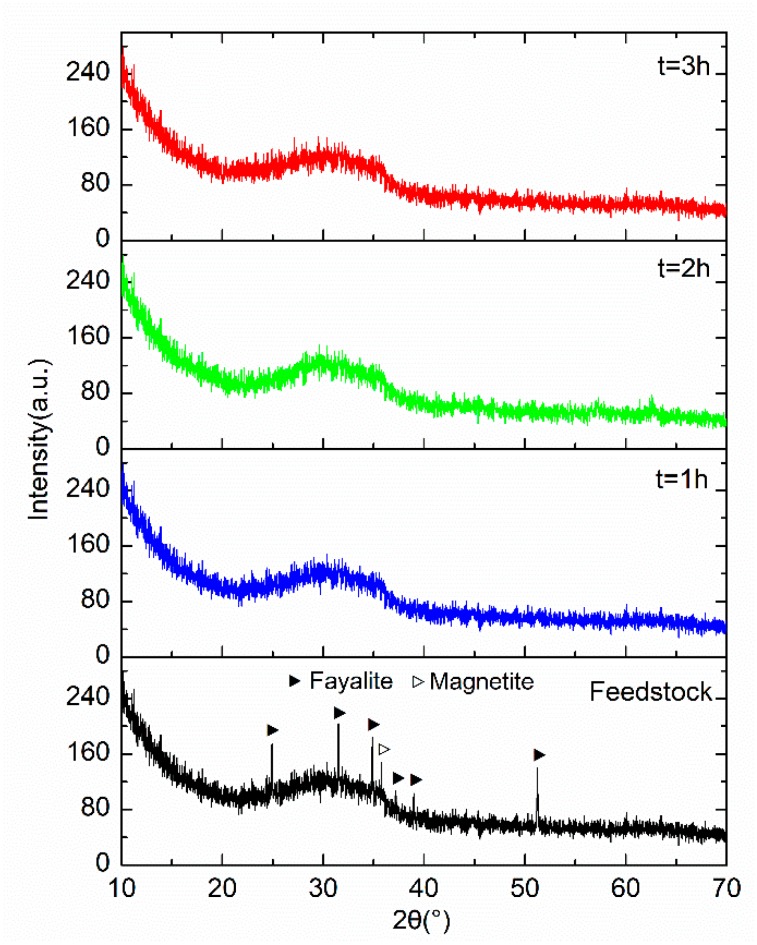
XRD patterns of GCS milled in a vibratory mill for different durations.

**Figure 2 materials-12-00772-f002:**
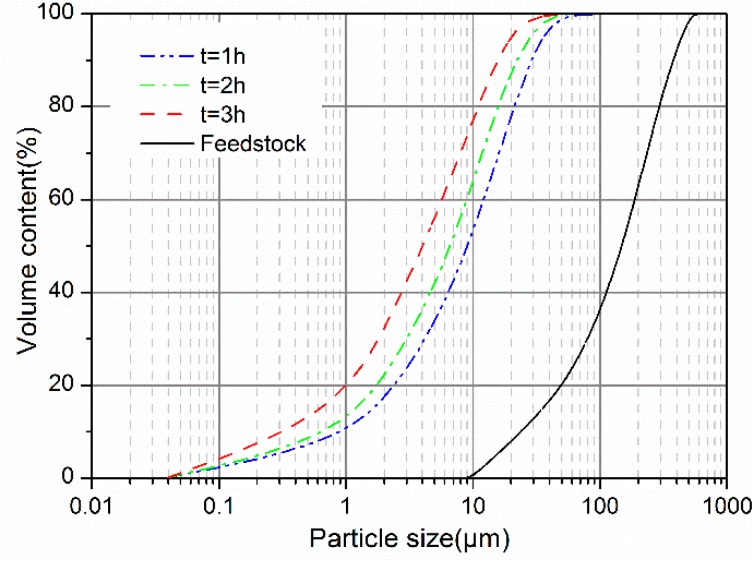
Particle size cumulative distribution of GCS as function of milling durations.

**Figure 3 materials-12-00772-f003:**
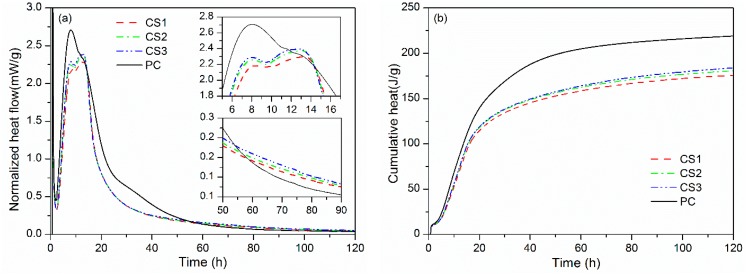
Isothermal calorimetry results for binders at 25 °C: (**a**) Normalized heat flow and (**b**) Cumulative heat.

**Figure 4 materials-12-00772-f004:**
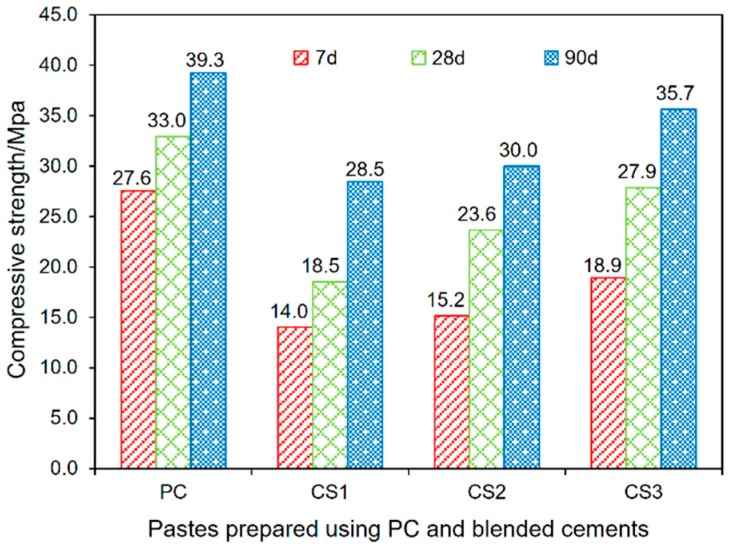
Compressive strength development of all tested pastes at 7, 28 and 90 days.

**Figure 5 materials-12-00772-f005:**
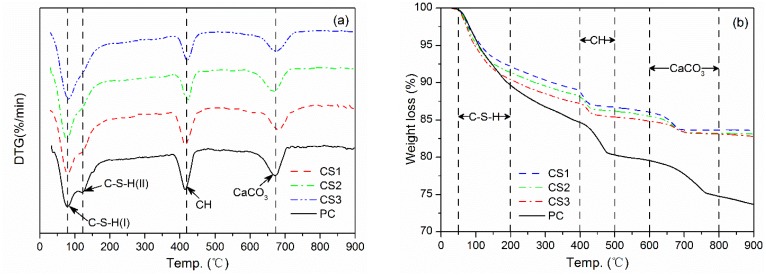
DTG/TGA curves of all pastes for 28 days of curing: (**a**) DTG curves; (**b**) TGA curves.

**Figure 6 materials-12-00772-f006:**
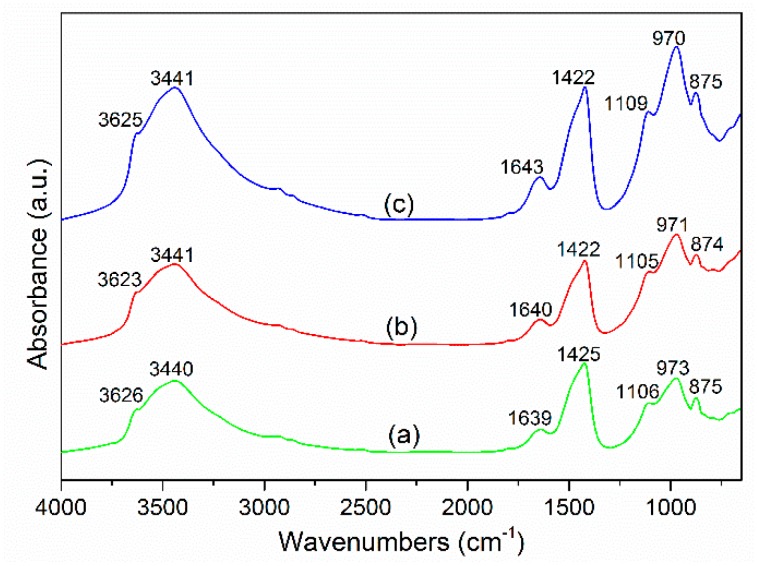
FTIR spectra of binder pastes for 28 days of curing at ambient temperature: (**a**) CS1, (**b**) CS2 and (**c**) CS3.

**Figure 7 materials-12-00772-f007:**
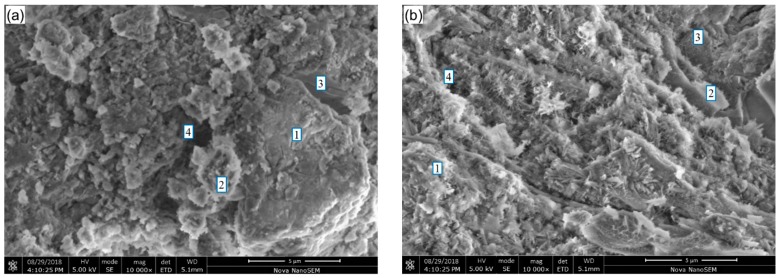
SEM images of binder pastes for 28 days of curing: (**a**) CS1, (**b**) CS2 and (**c**) CS3. 1. C–S–H; 2. Unreacted GCS; 3. CH; 4. Pore.

**Table 1 materials-12-00772-t001:** Chemical composition (%) of GCS and PC.

Material	SiO_2_	FeO	Fe_2_O_3_	CaO	Al_2_O_3_	MgO	Zn	K_2_O	SO_3_
GCS	33.40	35.89	7.14	4.00	3.50	1.39	1.08	–	–
PC	18.10	–	2.80	62.10	4.90	1.20	–	1.20	3.70

**Table 2 materials-12-00772-t002:** Characteristic particle sizes and specific surface area of GCS milled for various durations.

Milling Time	D10 (μm)	D50 (μm)	D90 (μm)	Specific Surface Area (m^2^/g)
1 h	1.2	9.4	27.2	0.67
2 h	1.0	6.6	20.0	1.03
3 h	0.7	4.1	14.2	1.37

**Table 3 materials-12-00772-t003:** Characteristic values of normalized heat flow and total heat emission of binders and PC.

Sample	Ending Time of the Induction Period (h)	Peak Value (mW/g)		Total Heat Emission (J/g)			
		The Second Peak	The Third Peak	12 h	24 h	48 h	120 h
CS1	2.8	2.2	2.3	68.5	124.4	151.5	175.8
CS2	2.6	2.3	2.4	71.8	127.7	155.0	180.9
CS3	2.5	2.3	2.4	72.8	128.5	156.2	183.9
PC	2.3	2.7	–	85.8	153.6	196.8	219.0

**Table 4 materials-12-00772-t004:** The ratio and strength development rate of compressive strength for all tested pastes.

Sample	Strength Ratio (%)			Strength Development Rate (MPa/d)	
	7 d	28 d	90 d	from 7 to 28 d	from 28 to 90 d
PC	100	100	100	0.26	0.10
CS1	51	56	72	0.21	0.16
CS2	55	72	76	0.40	0.10
CS3	69	84	91	0.43	0.13

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
