# Peer review of "Mechanical Activation of Granulated Copper Slag and Its Influence on Hydration Heat and Compressive Strength of Blended Cement"

_materials, 2019, doi:10.3390/ma12050772_

Reviewer 1 Report

This paper presents an experimental investigation of the production and evaluation of cement paste containing 30% GCS. The following comments are suggested:

1.    The title does not reflect the content of the article well.

2.    The introduction contains a lot of detailed literature information, but without any attempt to synthesize and analyze it.

3.    Abstract and conclusions need to be rewritten to report about the main and new findings obtained in this paper briefly. Particularly, the conclusions only general narrative.

4.    Please describe the basis for the determination of experimental parameters? Ex, 30% GCS, W/B=0.5.5.    On the XRD spectrum charts, please specify the number of counts on the ordinate axes. Can't tell the difference from the chart.

Author Response

Dear reviewer:

We are deeply grateful to thoughtful comments for the manuscript (materials-454269). Those comments are all valuable and very helpful for revising and improving our article, as well as the important guiding to our further researches.

We have made corrections and revisions using red text in the coverletter and revised manuscript according to the recommendations. The detailed response has been attached.

Best regards!

Reviewer 2 Report

Dear Editor,

Thanks for your invitation to review the article materials-442852 for your journal.

Please find my comments as follows:

Accept with minor amendments. The Manuscript is well written, the introduction provides sufficient state of art about mechanical activation of waste materials from industry.  

Extensive experimental work has been conducted by the authors to test the mechanically activated GCS.

The conclusion properly presents what has been done and what has been achieved through the Manuscript.

The only thing I suggest is to provide recommendations for further investigation and developments. Also, better care should be taken to remove error references from the Manuscript.

Regards

Author Response

(The authors gave the same response as above.)

Reviewer 3 Report

The subject taken in the article is interesting and current. Generally, the presented research has addressed a cognitively significant and applicationally problem. The research was carried out reliably, using various complementary test methods, the conclusions from the tests, although they are not surprising, are formulated correctly and result from the conducted research. Its serious drawback is the narrow scope of research - 1 cement, 1 slag, 3 milling grades - which can actually be considered as preliminary research. Due to the scope of the reviewer has doubts whether this is material for publication in a journal of international scope. In the opinion of the reviewer, the publication is premature and should take place after a wider scope of research.

Detailed comments:·

The title is too wide - only compressive strength was tested.·

The lack of justification for the milling time - 1-3 hours - is a large time period, why so long?·

Lack of information about the variability of GCS characteristics - is there material with similarly stable properties like eg. GBFS? GBFS considerations should be included in the context of GCS - Are they similar to each other or not?. The question is of course - are GCS and GBFS similar in properties? There is a lot of errors in references in the reviewed text.·

The scope of research is too narrow.·

The discussion of the results is very complicated, it can and should be re-edited so that it is easier to receive. There are too many fragments of the type of a handbook, obvious to the reader-oriented (and it's hard to expect others in the case of a specialized article). This applies to the discussion of all the research results presented in the article.

Author Response

Dear reviewer:

We are deeply grateful to thoughtful comments for the manuscript (materials-454269). Those comments are all valuable and very helpful for revising and improving our article, as well as the important guiding to our further researches.

We have made corrections and revisions using red text in the coverletter and revised manuscript according to the recommendations. The detailed response has been attached.

Best regards!

Round  2

Reviewer 3 Report

I accept corrections